# Isolation and Characterization of H1 Subtype Swine Influenza Viruses Recently Circulating in China

**DOI:** 10.3390/v17020185

**Published:** 2025-01-27

**Authors:** Minghao Yan, Tianxin Ma, Xiaona Shi, Qin Chen, Luzhao Li, Bangfeng Xu, Xue Pan, Qiaoyang Teng, Chunxiu Yuan, Dawei Yan, Zhifei Zhang, Qinfang Liu, Zejun Li

**Affiliations:** Department of Avian Infectious Diseases, Shanghai Veterinary Research Institute, Chinese Academy of Agricultural Sciences, Shanghai 200241, China; m17519478956@163.com (M.Y.); matianxin12@163.com (T.M.); shixiaonare@163.com (X.S.); chenqin0928p@163.com (Q.C.); luzhao.li@wur.nl (L.L.); xubangfeng@shvri.ac.cn (B.X.); panxue@shvri.ac.cn (X.P.); tengqy@shvri.ac.cn (Q.T.); yuanchx@shvri.ac.cn (C.Y.); yandawei865@163.com (D.Y.); nzhangzhifei@163.com (Z.Z.)

**Keywords:** swine influenza virus, H1 subtype, reassortant, replication, potential threat

## Abstract

Pigs serve as a mixing vessel for influenza viruses and can independently promote the emergence of pandemic strains in humans. During our surveillance of pig populations from 2021 to 2023 in China, 11 H1 subtype swine influenza viruses (SIVs) were isolated. All viruses were reassortants, possessing internal genes of identical origins (PB2, PB1, PA, NP, M: pdm09/H1N1 origin, NS: North American triple reassortant origin). The H1N1 isolates were all the dominant G4 EA H1N1 viruses in China. Two H1N2 isolates carried early human pdm09/H1N1 HA genes, suggesting a possible pig-to-human transmission route. Mutations that dictate host range specificity were identified in all isolates, a phenomenon which may enhance the affinity to human receptors. These H1 subtype viruses effectively replicated both in vivo and in vitro without prior adaptation and exhibited different pathogenicity and growth characteristics. Some of the H1 viruses were even found to cause lethal infections in mice. Taken together, our study indicates that the H1 subtype SIVs recently circulating in China pose a potential threat to human health and emphasizes the importance of continuing to closely monitor their evolution and spread.

## 1. Introduction

The swine influenza, caused by the swine influenza virus (SIV), is a severe acute respiratory disease that not only has a significant economic impact on the livestock industry, but also poses a serious threat to public health [1,2]. The SIV belongs to the influenza A viruses (IAV) in the *Orthomyxoviridae* family, with a genome consisting of eight single-stranded negative-sense RNA segments [3]. Based on the antigenic differences of its hemagglutinin (HA) and neuraminidase (NA) molecules, IAV could be classified into 18 HA subtypes (H1–H18) and 11 NA subtypes (N1–N11), most of which spread among wild birds [4]. The presence of both avian α-2,3 and human α-2,6 sialic acid receptors in the pig respiratory tract allows avian influenza and human influenza viruses to infect pigs [5]. This susceptibility enables viruses from different sources to acquire adaptive mutations or undergo genetic reassortment within pigs, thereby generating novel viruses with pandemic potential [6,7]. Therefore, pigs are considered as “mixing vessels” for the emergence of new influenza viruses, playing a crucial role in the transmission and evolution of influenza viruses [8].

Various human and avian influenza virus subtypes have been isolated from pigs, such as H1, H2, H3, H4, H5, H7, and H9 subtypes [9,10,11]. However, only three major subtypes of influenza viruses, H1N1, H1N2, and H3N2, have established themselves and spread among swine populations globally, a phenomenon which is strikingly similar to the epidemic subtypes in humans in the past and present [12,13]. The swine H1N1 influenza virus (known as classical swine H1N1) was initially identified during the H1N1 pandemic of 1918 and first isolated in the 1930s [14], with its isolation from humans reported in 1974 [15]. Shortly after the 1968 human pandemic, H3N2 SIV was identified in pigs in Taiwan, showing antigenic similarity with certain human strains [16]. Subsequent H1N2 viruses also emerged, with the first being discovered in Japanese pig populations in 1978, originating from a recombination between CS H1N1 and human H3N2 viruses, and quickly spread to other regions [17]. The SIV had been spreading at a rapid pace within pig populations until the late 1990s when a triple-reassortant gene (TRIG) emerged in new H3N2 viruses in North America containing gene segments from human, avian, and swine viruses [18]. The introduction of reassorted H3N2 influenza viruses accelerated the development of genetic diversity among influenza viruses in pigs [19]. The outbreak of the 2009 H1N1 influenza pandemic virus (pdm09/H1N1) caused a global pandemic among humans [20]. Following the pandemic, its reintroduction into pigs further enriched the genetic diversity of swine influenza viruses and altered the genomic landscape of IAV [21,22].

The triple-reassortant (TR) and Eurasian avian-like (EA) viruses co-circulate in the swine population in China [23]. The introduction of the pdm09/H1N1 virus has compounded this complexity, as its internal gene constellation has been evolving and frequently reassorting with various subtypes of the SIV [24]. This has led to the emergence of numerous new genotypes carrying the pdm09/H1N1 gene constellation [25]. These recombinant viruses carrying pdm09/H1N1 gene segments exhibit the hallmark characteristics of candidate pandemic viruses, significantly upping their adaptability to humans and posing a serious threat to human health [26,27]. Several cases of human infections with the SIV have been reported domestically [28,29]; although they have not caused a pandemic, they have raised concerns about the potential infectivity of these pdm09-reassorted SIV strains to humans.

The host’s innate immune defenses are insufficient to overcome SIV infection; thus, the current prevention and control of the swine influenza virus primarily rely on vaccination. The ongoing evolution of influenza viruses, the high mutation rates, and the genetic reassortment has spawned novel strains with markedly different antigenicity [3,30]. Although commercial vaccines are available, their effectiveness is largely constrained by the degree of antigenic match between the vaccine strains and the circulating virus strains. In China, the coexistence of different subtypes and genotypes within the same subtype of SIV adds another layer of complexity to the situation, making prevention and control efforts more challenging [31,32]. Therefore, continuous monitoring of the prevalence of the SIV in China is of vital importance.

To this end, our study procured nasal swabs from pigs in selected regions of China between 2021 and 2023 and isolated a total of 11 strains of the H1 subtype of the SIV. Through whole genome sequencing, we performed a systematic analysis of the phylogenetic ties and molecular characterization of the isolated strains. Additionally, the replication of the H1 subtype of the SIV in mammalian cells was tested in vitro, and mice were used as infection animal models to assess the virus’s pathogenicity in mammals. These results provide important information for understanding the genetic and evolutionary characteristics of the H1 subtype of SIVs recently circulating in China, and it is recommended to closely monitor and assess their potential threat to public health.

## 2. Materials and Methods

### 2.1. Sample Collection and Virus Isolation

From June 2021 to November 2023, we collected 953 nasal swab samples from pigs with symptoms typical of the swine influenza on pig farms across eight provinces (Shanghai, Jiangsu, Zhejiang, Shandong, Beijing, Henan, Sichuan, and Inner Mongolia) in China. Samples were collected in sterile phosphate-buffered saline (PBS) containing antibiotics and transported to the laboratory at low temperatures (−20–−70 °C). Before use, the nasal swabs were naturally thawed on ice and centrifuged at 3000 rpm for 10 min at 4 °C to separate the supernatant. The supernatant was then filtered through a filter with a 0.22 μm pore size for inoculation onto Madin–Darby canine kidney (MDCK) cells in six-well plates. The cells were cultured at 37 °C for 72 h in an SFM medium containing tosylsulfonyl phenylalanyl chloromethyl ketone (TPCK)-trypsin (2 μg/mL). After three blind passages, the supernatant was collected for the hemagglutination (HA) assay. Positive supernatants were stored at −80 °C.

### 2.2. RT-PCR and Sequencing

A reverse transcription–polymerase chain reaction (RT-PCR) was performed to amplify the viral RNA for sequencing and phylogenetic analysis. Viral RNA was extracted using the TRIZOL reagent (AG accurate Biology, Changsha, China) according to the standard protocol. The isolated RNA was reverse transcribed into cDNA using M-MLV reverse transcriptase (Vazyme, Nanjing, China) and the influenza A virus Uni 12 primer (AGCAAAAGCAGG). PCR was performed using segment-specific primers for eight genes (PB2, PB1, PA, HA, NP, NA, M, and NS), as reported previously (Hoffmann et al., 2001). The viral gene segments were sequenced by the Shanghai Qingke Biotechnology Co., Ltd. (Shanghai, China).

### 2.3. Sequence Analyses

Reference sequences of human, swine, and avian IAVs were obtained from the Influenza Virus Resource of the NCBI (http://www.ncbi.nlm.nih.gov/genomes/FLU/FLU.html, accessed on 9 March 2024). DNA sequences were aligned and analyzed using the MegAlign and Editseq programs in Lasergene 7.1 (DNASTAR, Madison, WI, USA). Phylogenetic trees were generated by the distance-based neighbor-joining method using the software MEGA 6.0 (Sinauer Associates, Inc., Sunderland, MA, USA). The reliability of the tree was assessed by bootstrap analysis with 1000 replicates (Saitou and Nei, 1987). The nucleotide regions used in the phylogenic analysis were as follows: H1 HA, nt 33–1733; N1 NA, nt 21–1430; N2 NA, nt 20–1429; PB2, nt 28–2307; PB1, nt 25–2298; PA, nt 25–2175; NP, nt 46–1542; M, nt 26–1007; NS, nt 27–864.

### 2.4. Virtual Replication In Vitro

To assess the in vitro growth characteristics of the virus, MDCK cells were infected with each virus at a multiplicity of infection (MOI) of 0.001. After incubation at 37 °C for 1 h, the cells were washed twice with PBS and further incubated in the SFM medium containing TPCK. Culture supernatants were harvested 12, 24, 48, and 72 h post-inoculation (hpi) and stored at −80 °C. The viral titers were determined by endpoint titration in the MDCK cells and expressed as the mean log_10_ TCID_50_/_mL_ ± standard deviation (SD).

### 2.5. Mouse Experiments

To evaluate the pathogenicity of isolates in mammalian hosts, four-to-five-week-old SPF female BALB/c mice were randomly divided into four groups of 11 mice each. The mice were inoculated intranasally with 10^6^ TCID_50_ of each virus in a 50 μL volume after anesthesia with CO_2_. Additionally, mice infected with 50 μL of PBS were used as negative controls. Three mice in each group were sacrificed 4 and 6 days post-infection (dpi), and tissues including the brain, nasal turbinates, lungs, spleen, and kidneys were collected and stored at −80 °C for virus titration. The remaining mice were monitored for 14 days to assess mortality, weight changes, and clinical symptoms. A weight loss of 25% or more was considered as death. Viral titers were evaluated by calculating the 50% tissue culture infectious dose (TCID_50_) using the Reed and Muench method in the MDCK cells.

### 2.6. Histopathology

The four-to-five-week-old SPF female BALB/c mice were inoculated intranasally with 50 μL (10^6^ TCID_50_) of each virus. Three mice in each group were sacrificed at 4 dpi. While collecting organs for virus titration, half of the infected mouse lung tissue samples were taken out and immersed in a 10% neutral buffered formalin solution, processed, and embedded in paraffin. Lung sections were cut into 4 μm samples and stained with hematoxylin and eosin (H&E) for histopathological examination.

### 2.7. Ethics Statement

The four-to-five-week-old SPF female BALB/c mice were purchased from Beijing Vital River Laboratory Animal Technology Co., Ltd., Beijing, China. All animal studies in this research were conducted in accordance with the guidelines of the Animal Care and Use Committee of the Shanghai Veterinary Research Institute, and all animal study protocols were approved by the Chinese Academy of Agricultural Science (permit number: SHVRI-Po-0102).

## 3. Results

### 3.1. Virus Isolation and Identification

In an effort to assess the epidemiological status of SIVs, from June 2021 to November 2023, a total of 953 swine nasal swabs were collected from pig farms in eight provinces of China, including Shanghai, Jiangsu, Zhejiang, Shandong, Beijing, Henan, Sichuan, and Inner Mongolia. Virus isolation was performed using MDCK cells, followed by hemagglutination assays for identification and by subsequent confirmation through RT-PCR and genomic sequencing. This process, ultimately, resulted in the successful isolation of 11 SIV strains, comprising nine strains of the H1N1 subtype and two strains of the H1N2 subtype.

### 3.2. Phylogenetic Analysis

To systematically analyze the genetic evolution of SIVs in China from 2021 to 2023, sequencing was performed on all eight genes of each virus, followed by phylogenetic analyses. Nine phylogenetic trees were constructed using nucleotide sequences of representative viruses available in the GenBank database.

Phylogenetic analysis of the HA gene indicated that all 11 isolates found in this study belonged to the H1 subtype and were located in two different lineages. Among them, nine isolates belonged to the EA H1N1 lineage, while two belonged to the pdm09/H1N1 lineage. The nucleotide sequence identity among the 11 H1 genes ranged from 69.6% to 99.9% (Figure 1A). Regarding the NA genes, all the nine N1 NA genes had a nucleotide sequence homology of 92.7–100% and clustered in the EA H1N1 lineage (Figure 1B). Two N2 NA genes clustered into the TR-H1N2 lineage, with a nucleotide sequence identity of 94.7% (Figure 1C).

Unlike the diversity observed in the HA and NA genes, all six internal genes of the isolates in this study belonged to the same lineage (Figure 2A–F). The polymerase basic protein 2 (PB2), polymerase basic protein 1 (PB1), polymerase acidic protein (PA), nucleoprotein (NP), and matrix protein (M) genes of the 11 isolates all clustered in the pdm09/H1N1 lineage, with nucleotide similarities of 95.0–100%, 95.3–100%, 94.1–100%, 93.3–99.9%, and 94.7–100%, respectively. The NS genes of the isolates were all classified into the lineage of TR-H1N2 and shared a 95.4–100% identity at the nucleotide level.

### 3.3. Genotyping Analysis

The rearranged genotype of the isolated strains was defined based on the phylogenetic analysis of the eight gene segments. All isolates harbored PB2, PB1, PA, NP, and M gene segments derived from the pdm/09 H1N1 lineage and NS gene segments derived from the TR H1N2 lineage. The nine H1N1 strains had HA and NA gene segments originating from the EA H1N1 lineage, which belongs to the predominant triple-reassortant G4 EA H1N1 virus currently circulating in China. The two H1N2 strains not only possessed five internal genes from the pdm/09 H1N1 lineage, but also inherited the HA gene from the pdm/09 H1N1 lineage. They were dual-reassortant viruses that had arisen from the reassortment of pdm/09 H1N1 and TR H1N2 (Figure 3).

### 3.4. Molecular Characterization of the Viruses

Based on the sequence analysis, we compared the key molecular characteristics related to the receptor binding ability, drug resistance, and virulence of these SIVs.

HA comprises two subunits, HA1 and HA2, which are covalently linked by a disulfide bond [33]. All of our isolated SIVs had a single basic amino acid, PSIQSR/G, at the HA cleavage site (Table 1), consistent with the characteristics of low pathogenic avian IAVs [34]. N-linked glycosylation is crucial for protein folding and maturation [35]. The HA protein from all H1 subtype viruses had five conserved potential N-glycosylation sites at positions 13, 14, 23, 484, and 543 (H3 numbering). Nine EA H1N1 viruses had glycosylation sites at positions 198 and 277; however, two H1N2 viruses and their genetic ancestor A/California/04/09 (CA09) had glycosylation sites at positions 90, 279, and 290.

It is generally accepted that a receptor-binding preference for human-type receptors is the initial key step for a novel influenza virus causing a pandemic. All HA genes of the EA H1N1 and the pandemic-like H1N2 SIVs exhibited the E190D and G225E/D mutations (Table 1), both of which increase the receptor-binding affinity to human-type α-2,6-linked sialic acid receptors [36,37], suggesting that the virus may have potential to infect humans.

Two classes of antiviral drugs, M2 ion channel inhibitors and neuraminidase inhibitors (NAIs), have been approved for the treatment of influenza virus infections. No isolates had NAI-resistant amino acids such as H274Y in the N1 gene and E119V and R292K in the N2 gene [38], indicating that these viruses retained a high sensitivity to NAIs. However, some H1 isolates harbored the V27I mutation and all H1 isolates harbored the S31N mutation (Table 1), conferring amantadine resistance [39].

The 627K and 701N amino acid residues in PB2 are now considered determinants of virulence and pathogenicity in several mammalian species [40,41]. All H1 subtype SIVs contained K at position 627 and D at position 701 with no mutations. However, we detected the T271A-A590S-A591R combination mutation, which is known to play a critical role in the replication and virulence of SIVs in mammalian hosts [42]. The presence of 251K, 431M, and 588I was also found in the isolates, increasing viral polymerase activity [43,44,45]. The P42S mutation and the K186E mutation were found in the NS gene of all viruses, suggesting that these viruses might exhibit greater resistance to antiviral cytokines [46,47]. Meanwhile, the Q357K mutation in the NP gene was observed (Table 1), a mutation which enhances the adaptation of swine influenza viruses to host populations and increases the pathogenicity in mice [48].

### 3.5. Growth Kinetics of H1 Subtype Viruses in Mammalian Cells

To evaluate the in vitro replication ability of the H1 subtype SIV isolates, the growth kinetics of various SIVs were examined in MDCK cells. The results demonstrate that viruses in this study replicated efficiently within MDCK cells. Specifically, the peak of virus replication occurred between 48 and 72 h post-inoculation (hpi), with average peaking titers varying from 6.08 to 8.25 log_10_ TCID_50_/mL. Some EA H1N1 SIVs displayed replication kinetics similar to those of the CA09 virus; however, two pandemic-like H1N2 viruses, JS/235/23 (H1N2) and ZJ/284/23 (H1N2), showed lower viral titers (Figure 4).

### 3.6. Replication and Pathogenicity of H1 Subtype Viruses in Mice

In general, mice are widely recognized as a valuable animal model for evaluating the virulence of influenza viruses to humans [49]. Our study evaluated the replication and pathogenicity of nine H1 subtype viruses in mice and compared them with the H1N1pdm09 virus (CA09). In our study, 4–5-week-old SPF female BALB/c mice were inoculated with a viral dose of 10^6^ TCID_50_ per virus and were monitored for 14 days for signs of illness, weight loss, and mortality. Compared with the uninfected group, all mice infected with H1 subtype SIVs, except for SD/2684/23 (H1N1), exhibited varying degrees of weight loss within 8 dpi, along with clinical symptoms such as lethargy, ruffled fur, huddling, and shivering (Figure 5A). Mice infected with SC/F1/22 (H1N1), NM/F1/23 (H1N1), and JS/235/23 (H1N2) gradually regained weight and their clinical symptoms subsided in the later stages of infection. One to two mice experienced a weight loss of over 25% and were, therefore, considered dead during the post-infection observation period. However, mice inoculated with BJ/F3/22 (H1N1), JS/F2/21 (H1N1), JS/02/21 (H1N1), and SH/2/22 (H1N1) experienced 100% mortality between days 4 dpi and 9 dpi, comparable to the highly pathogenic CA09 virus (Figure 5B).

At 4 and 6 dpi, three mice from each group were randomly sacrificed, and tissues were collected to determine viral loads. No virus was detected in the brain, spleen, or kidneys of any infected mice. However, all nine viruses effectively replicated in the lungs and nasal turbinates. Except for SD/2684/23 (H1N1) and ZJ/284/23 (H1N2), which showed lower replication levels in the nasal turbinates, the other H1 subtype viruses exhibited a similar in vivo replication ability to CA09, with high titers in both lungs and nasal turbinates (Figure 5C,D). These results indicate that these reassortant H1 subtype SIVs preferentially infected the respiratory system, rather than extra-pulmonary tissues, without prior adaptation.

### 3.7. Histopathological Damage to Mouse Lungs

To establish a correlation between virulence and pathogenicity, lung tissues from infected mice were collected at 4 dpi and stained with hematoxylin and eosin (H&E). An observational analysis revealed that, compared to the lungs of the PBS-inoculated control mice, mice infected with all the viruses exhibited varying degrees of histopathological damage in the lungs, presenting typical lesions of influenza A virus infection: bronchiolitis with respiratory epithelial necrosis and alveolitis. However, the severity and characteristics of necrosis and inflammation varied slightly depending on the strain. The virus with the lowest virulence, SD/2684/23 (H1N1), exhibited mild alveolitis with increased infiltration of inflammatory cells but no necrosis. In contrast, other viruses caused more severe pathological changes, characterized by significant disruption of alveolar structures, extensive necrosis and shedding of bronchiolar epithelial cells, and infiltration of inflammatory cells. Within the alveolar spaces, a modest number of lymphocytes, macrophages, and neutrophils were observed engulfing cellular debris from the necrotic areas (Figure 6).

## 4. Discussion

Currently, pigs worldwide have been found to harbor a range of influenza A viruses, including avian/human, human/swine, and human/avian/swine strains [50]. China stands out as having the most diverse ecosystem of SIVs. Since the emergence of the 2009 pandemic H1N1 viruses (pdm/09), there has been frequent reassortment with local SIVs, some of which acquired the ability to infect humans [51,52]. Our recent surveillance study conducted from 2021 to 2023 identified 11 H1 subtype strains, including nine H1N1 (81.82%) and two H1N2 (18.18%) subtypes. The H1N1 SIV remains the predominant circulating subtype in China.

Influenza viruses produce different subtypes or genotypes by exchanging gene segments through recombination, leading to the emergence of new epidemic strains [53]. In our study, all H1 subtype isolates possessed the same internal gene constellation—containing the TR H1N2 NS genes with the remaining five genes originating from pdm09/H1N1, which may represent an advantageous selection for the SIVs that have recently been circulating in China as they adapt to their pig hosts. The H1N1 subtype strains all belonged to the most widely circulating G4 EA H1N1 viruses in China which had already acquired strong characteristics for infecting humans [26]. The high compatibility of the internal genes of pdm09/H1N1 with different surface genes gives it a significant competitive advantage in pigs. In contrast to the enduring presence of internal genes from the pdm09 lineage, the pdm09-origin surface genes are not sufficiently competitive with the enzootic SIVs, leading to a lack of persistence of pdm09/H1N1 viruses and their surface genes in pigs [25]. However, the two H1N2 subtype strains isolated in this study retained the HA gene specific to the early pdm09/H1N1 virus. This early gene was significantly different from the currently used vaccine antigen, a phenomenon which may impact the effectiveness of existing vaccines against these circulating strains.

Species barriers prevent the influenza virus from infecting one host from another. The influenza virus must overcome this host range barrier to effectively infect and spread [54]. Normally, human influenza viruses prefer to bind to α-2,6 receptors, while avian influenza viruses have a greater preference for α-2,3 receptors. Therefore, it is generally accepted that an HA receptor-binding preference for a-2,6-linked sialylated glycans is the initial key step for a novel influenza virus causing a pandemic [55]. In our study, all H1 subtype viruses exhibited the E190D and G225D/E mutations. These mutations confer the ability to bind to human-type receptors with high affinity [36,37], indicating that the virus may have the potential to infect humans.

In recent years, concerns regarding low-pathogenic influenza have grown, as it may be able to replicate efficiently in vitro and in vivo and have the ability to infect humans. Our study found that all H1 subtype viruses were able to replicate effectively in MDCK cells without the need for prior adaptation. The combination of 271A, 590S, and 591R is known to be crucial for the efficient replication and adaptation of SIVs in cultured cells [42]. This genetic polymorphism was confirmed in our isolates, which provided the conditions for the virus to replicate effectively in the cells. The pathogenicity of the viruses in mice was assessed, and the results indicate that the circulating SIV strains carrying avian-sourced HA genes typically exhibit higher virulence. Mice infected with four H1N1 subtype strains (BJ/F3/22, JS/F2/21, JS/02/21, and SH/2/22) experienced rapid weight loss, severe illness, and 100% mortality. Infections with other H1 subtype viruses resulted in moderate weight loss and partial mortality in mice. The R251K substitution in PB2 increases viral replication and pathogenicity of EA H1N1 SIVs [44]. These low-virulence mutations may have been the reason for the lower mortality rates observed in mice infected with certain EA H1N1 viruses. The two H1N2 viruses, JS/235/23 (H1N2) and ZJ/284/23 (H1N2), demonstrated similar replication capabilities in vivo to their genetic ancestors, yet there was a significant difference in their pathogenicity. Compared to highly pathogenic viruses, the covert transmission characteristics of these viruses with low virulence and high replication levels may increase their risk of adaptation and spread in humans, making them more difficult to detect and control in a timely manner. Interestingly, none of the viruses caused systemic infections in mice, and the viruses effectively replicated only in the lungs and nasal turbinates of the mice. Significant histopathological damage was observed in lungs with high viral loads, marked by widespread epithelial cell necrosis and recruitment of inflammatory cells.

In summary, our findings suggest that the H1 subtype swine influenza viruses which have been recently circulating in China demonstrate efficient replication capabilities in mammalian hosts. It is challenging to predict whether, through additional adaptation and reassortment, they might evolve into viruses capable of efficient transmission among human populations in the future. Given the potential public health threat posed by these circulating SIVs, it is essential to conduct continuous surveillance and genetic characterization to inform vaccine strain selection and update strategies.

## Figures and Tables

**Figure 1 viruses-17-00185-f001:**
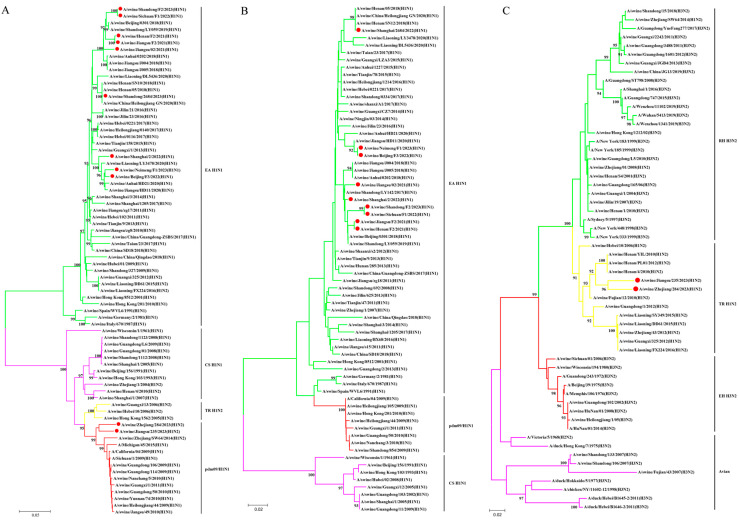
Phylogenetic relationships of HA gene and NA gene segments of the H1 subtype of SIVs in China in the period from 2021 to 2023. (**A**) H1-HA gene. (**B**) N1-NA gene. (**C**) N2-NA gene. Phylogenetic trees were constructed using the distance-based neighbor-joining method with the maximum composite likelihood model and MEGA 6.0 with 1000 bootstrap replicates. The scale bar indicates the number of nucleotide substitutions per site. pdm09/H1N1, 2009 pandemic H1N1; EA H1N1, Eurasian avian-like H1N1; CS H1N1, classical swine H1N1; TR H1N2, triple reassortant H1N2; RH H3N2, recent human H3N2; EH H3N2, early human H3N2. The 11 new SIV isolates from this study are marked with red dots.

**Figure 2 viruses-17-00185-f002:**
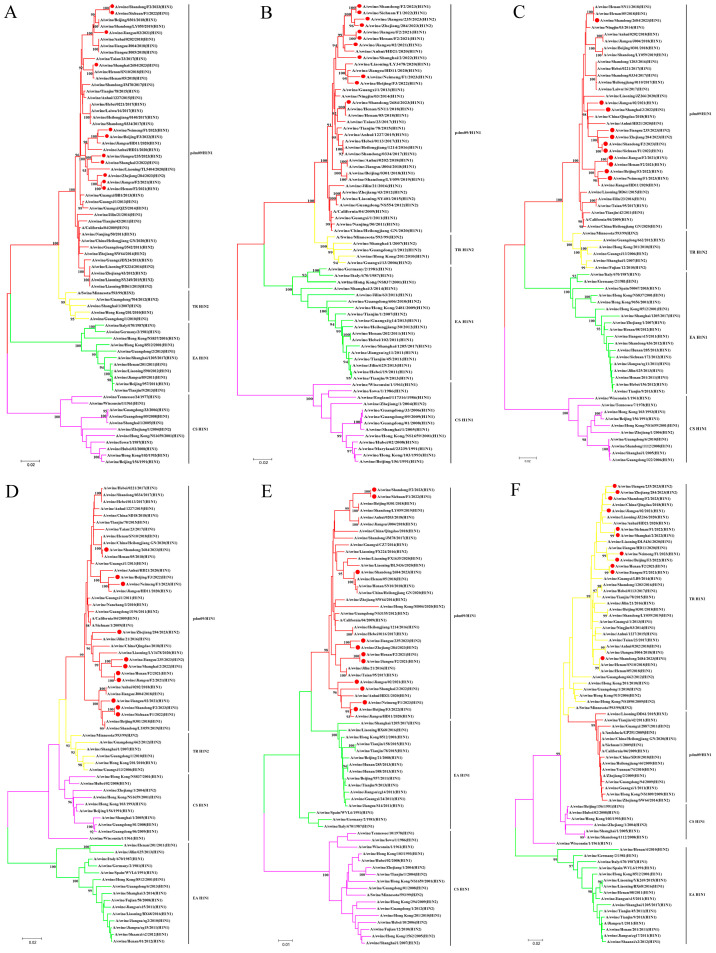
Phylogenetic relationships of internal gene segments of the H1 subtype of SIVs in China in the period from 2021 to 2023. (**A**) PB2 gene. (**B**) PB1 gene. (**C**) PA gene. (**D**) NP gene. (**E**) M gene. (**F**) NS gene. Phylogenetic trees were constructed using the distance-based neighbor-joining method with the maximum composite likelihood model and MEGA 6.0 with 1000 bootstrap replicates. The scale bar indicates the number of nucleotide substitutions per site. pdm09/H1N1, 2009 pandemic H1N1; EA H1N1, Eurasian avian-like H1N1; CS H1N1, classical swine H1N1; TR H1N2, triple reassortant H1N2. The 11 new SIV isolates from this study are marked with red dots.

**Figure 3 viruses-17-00185-f003:**
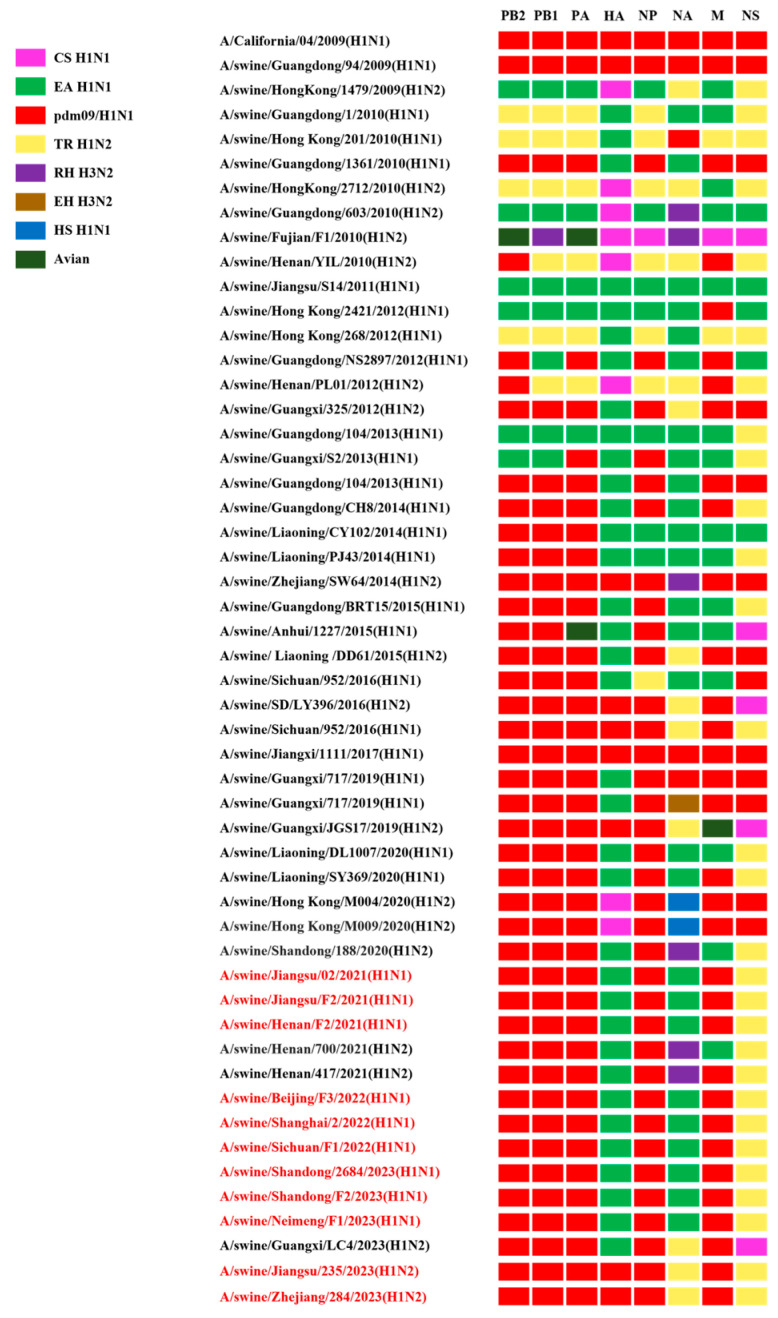
The genotypes of the H1 subtype isolates. All eight gene segments of SIVs are at the top of each bar. The colors of the bars indicate the lineage origin of the gene segments. pdm09/H1N1, 2009 pandemic H1N1; EA H1N1, Eurasian avian-like H1N1; TR H1N2, triple reassortant H1N2; RH H3N2, recent human H3N2; EH H3N2, early human H3N2; HS H1N1, human seasonal H1N1.

**Figure 4 viruses-17-00185-f004:**
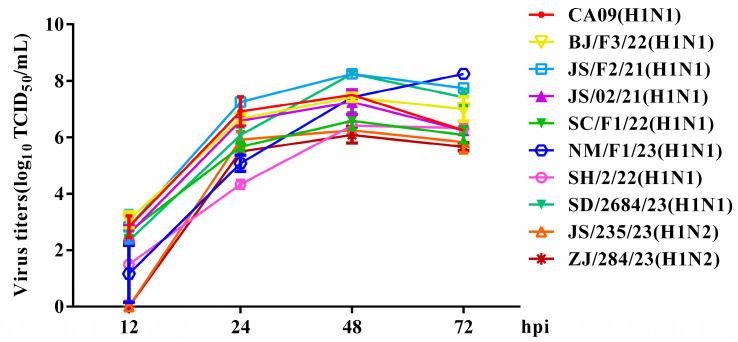
Growth kinetics of the isolated SIVs in mammalian cells. MDCK cells were infected with equal amounts of viruses at an MOI of 0.001. The cell supernatant was harvested at 12, 24, 48 and 72 hpi and titrated in MDCK cells. The data shown are the mean virus titers ± SD in log_10_ TCID_50_/mL.

**Figure 5 viruses-17-00185-f005:**
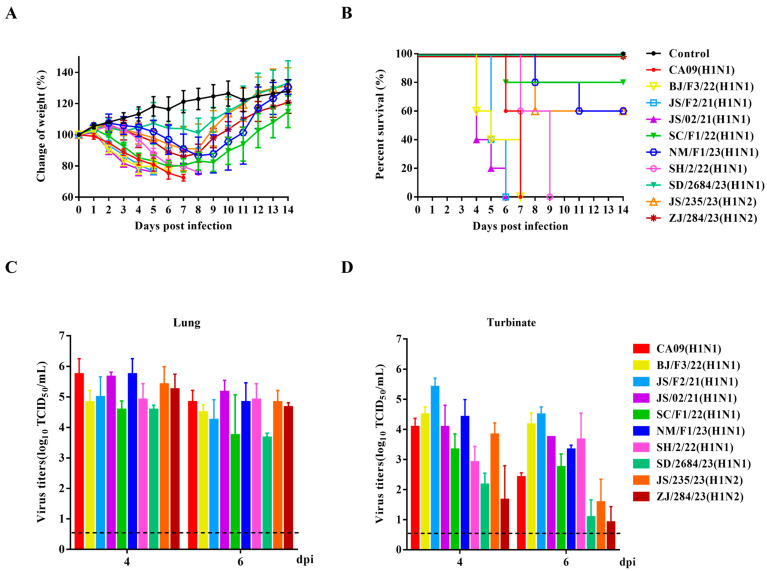
Pathogenicity of the isolated SIVs in mice. Four-to-five-week-old SPF female BALB/c mice (*n* = 5 per group) were inoculated intranasally with corresponding viruses at a dose of 10^6^ TCID_50_. Mice receiving an equal volume of PBS served as negative controls. The body weight changes (**A**) and survival rates (**B**) of mice were monitored and recorded daily for 14 consecutive days. The virus titers in the lungs (**C**) and nasal turbinates (**D**) of each mouse on day 4 and day 6 were determined by using end-point titration in the MDCK cells. Virus titers wareere expressed as log_10_ TCID_50_/mL. Data are presented as the mean ± SD.

**Figure 6 viruses-17-00185-f006:**
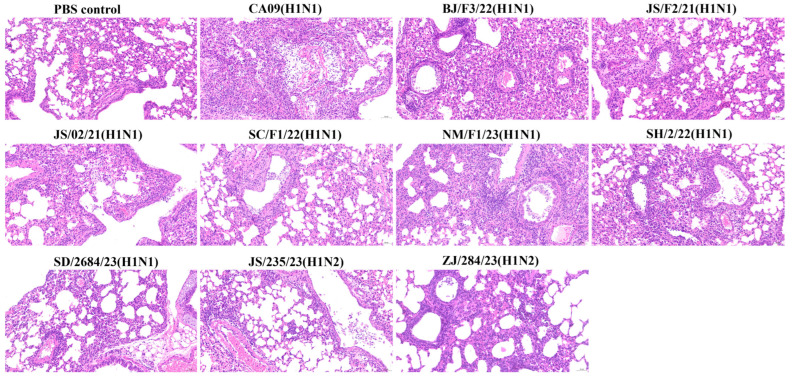
Pathological changes in the lungs of mice inoculated with the viruses. Mice were sacrificed at 4 dpi with 10^6^ TCID_50_ of the corresponding virus. Mice receiving PBS were used as controls. Lung tissues were fixed in formalin, stained with hematoxylin and eosin (H&E), and observed under a microscope at 200× magnification.

**Table 1 viruses-17-00185-t001:** Key amino acid sites in reassortant H1 isolates compared with the CA09 isolate.

	Cleavage Site	Mutation in HA That Alters the Affinity to Human-Type Receptors		Mutations in Different Genes That Alter Resistance to Antiviral Drugs					Mutations in Different Genes that Alter the Replication or Virulence										
Virus		HA ^a^		NA ^b^			M		PB2								NS		NP
		190	225	119	292	274	27	31	251	271	431	588	590	591	627	701	42	186	357
CA09 (H1N1)	PSIQSR/GLF	D	D			H	V	N	R	A	M	T	S	R	E	D	S	E	K
JS/F2/21 (H1N1)	PSIQSR/GLF	D	E			H	I	N	K	A	M	I	S	R	E	D	S	E	K
JS/02/21 (H1N1)	PSIQSR/GLF	D	E			H	V	N	K	A	M	I	S	R	E	D	S	E	K
HN/F2/21 (H1N1)	PSIQSR/GLF	D	E			H	I	N	K	A	M	I	S	R	E	D	S	E	K
BJ/F3/22 (H1N1)	PSIQSR/GLF	D	E			H	V	N	K	A	M	I	S	R	E	D	S	E	K
SH/2/22 (H1N1)	PSIQSR/GLF	D	E			H	V	N	K	A	M	I	S	R	E	D	S	E	K
SC/F1/22 (H1N1)	PSIQSR/GLF	D	E			H	I	N	R	A	M	I	S	R	E	D	S	E	K
SD/2684/23 (H1N1)	PSIQSR/GLF	D	E			H	V	N	R	A	M	I	S	R	E	D	S	E	K
SD/F2/23 (H1N1)	PSIQSR/GLF	D	E			H	I	N	R	A	M	I	S	R	E	D	S	E	K
NM/F1/23 (H1N1)	PSIQSR/GLF	D	G			H	V	N	K	A	M	I	S	R	E	D	S	E	K
JS/235/23 (H1N2)	PSIQSR/GLF	D	D	E	R		I	N	R	A	M	I	S	R	E	D	S	E	K
ZJ/284/23 (H1N2)	PSIQSR/GLF	D	D	E	R		I	N	R	A	M	I	S	R	E	D	S	E	K

^a^ The amino acid numbering follows the H3 numbering system. ^b^ The amino acid numbering follows the N2 numbering system.

## Data Availability

The data that supports the findings of this study are available from the corresponding authors on request.

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
