# Peer review of "Isolation and Characterization of H1 Subtype Swine Influenza Viruses Recently Circulating in China"

_viruses, 2025, doi:10.3390/v17020185_

Round 1

Reviewer 1 Report

Comments and Suggestions for Authors

The authors have performed a systemic study to isolate and characterize H1 subtype Swine influenza viruses circulating from 2021 to 2023 in China. The study has emphasized the importance of surveillance and monitoring of swine influenza viruses (SIVs) due to their potential threat to human health. The study was designed and performed very well with sound scientific knowledge and methodologies. The manuscript was written logically, with an adequate introduction, experimental design, and comprehensive results. The results were supported very well using appropriate figures, tables, discussion and citations.  However, the sequencing method used in this study was not clear. The addition of SNP analysis and showing the SNP tree would strengthen the visualization of isolated SIV mutations. Overall, the study highlights the need to closely monitor the Swine influenza viruses to prevent potential health risks to humans by continuous surveillance using different technologies. The manuscript has grammatical and spelling mistakes and needs to be reviewed thoroughly before accepting for publication.

Below are my comments/corrections highlighted in a red color font for some of the grammatical and spelling mistakes

Line 2-3 Isolation and Characterization of H1 Subtype Swine Influenza 2 Viruses Recently Circlating circulating in China

 Line 18: All viruses were reassortants, possessing, replace replaced internally with internal genes of identical

Line 25-26: circlating  circulating

Line 31: The swine influenza caused by the swine influenza virus (SIV) is a severe acute

Line 42: Therefore, pigs are thus considered as "mixing vessels" for the emergence of new

Line 56: SIV had continued at a rapid pace within pig populations until the late 1990s when a triple-

Line 62-63: Following the pandemic, its reintroduction into pigs further enriched the genetic diversity of swine influenza viruses and altered the genomic landscape of IAV

Line 92: SIVs recently circlating circulating

Line 99-100: containing antibiotics, and transported to the laboratory at low temperatures. Please write the low-temperature range

Line 103: MDCK cells, did not see the full name of MDCK in the manuscript, Madin-Darby canine kidney (MDCK) cells

Line 182: Unlike the diversity observed in the HA and NA genes, all 6 internal genes, of the isolated in this study belonged to the same lineage (Figures 2A-F).

Line 225: HA comprises two subunits, HA1 and HA2, which are covalently linked by a disulfide bond[33]. A space needs to be created before these types of citations.

Line 233: It is generally accepted that receptor-binding preference to human-type receptor receptors is the initial key step for a novel influenza-virus-causing-pandemic causing pandemic.

Author Response

Comments and Suggestions for Authors

The authors have performed a systemic study to isolate and characterize H1 subtype Swine influenza viruses circulating from 2021 to 2023 in China. The study has emphasized the importance of surveillance and monitoring of swine influenza viruses (SIVs) due to their potential threat to human health. The study was designed and performed very well with sound scientific knowledge and methodologies. The manuscript was written logically, with an adequate introduction, experimental design, and comprehensive results. The results were supported very well using appropriate figures, tables, discussion and citations.  However, the sequencing method used in this study was not clear. The addition of SNP analysis and showing the SNP tree would strengthen the visualization of isolated SIV mutations. Overall, the study highlights the need to closely monitor the Swine influenza viruses to prevent potential health risks to humans by continuous surveillance using different technologies. The manuscript has grammatical and spelling mistakes and needs to be reviewed thoroughly before accepting for publication.

Below are my comments/corrections highlighted in a red color font for some of the grammatical and spelling mistakes

Thank you very much for taking the time to review this manuscript. Please find the detailed responses below and the corresponding revisions in the re-submitted files.

Comments 1: Line 2-3 Isolation and Characterization of H1 Subtype Swine Influenza 2 Viruses Recently Circlating circulating in China

Answer 1: Thank you for pointing this out. I agree with this comment. The sentence has been modified in the revised manuscript (page 1, line 2-3).

Comments 2: Line 18: All viruses were reassortants, possessing, replace replaced internally with internal genes of identical

Answer 2: Thank you for pointing this out. I agree with this comment. The sentence has been modified in the revised manuscript (page 1, line 17).

Comments 3: Line 25-26: circlating  circulating

Answer 3: Thank you for pointing this out. I agree with this comment. The sentence has been modified in the revised manuscript (page 1, line 24-25).

Comments 4: Line 31: The swine influenza caused by the swine influenza virus (SIV) is a severe acute

Answer 4: Thank you for pointing this out. I agree with this comment. The sentence has been modified in the revised manuscript (page 1, line 30).

Comments 5: Line 42: Therefore, pigs are thus considered as "mixing vessels" for the emergence of new

Answer 5: Thank you for pointing this out. I agree with this comment. The sentence has been modified in the revised manuscript (page 1, line 41).

Comments 6: Line 56: SIV had continued at a rapid pace within pig populations until the late 1990s when a triple-

Answer 6: Thank you for pointing this out. I agree with this comment. The sentence has been modified in the revised manuscript (page 2, line 55).

Comments 7: Line 62-63: Following the pandemic, its reintroduction into pigs further enriched the genetic diversity of swine influenza viruses and altered the genomic landscape of IAV

Answer 7: Thank you for pointing this out. I agree with this comment. The sentence has been modified in the revised manuscript (page 2, line 61-62).

Comments 8: Line 92: SIVs recently circlating circulating

Answer 8: Thank you for pointing this out. I agree with this comment. The sentence has been modified in the revised manuscript (page 2, line 91).

Comments 9: Line 99-100: containing antibiotics, and transported to the laboratory at low temperatures. Please write the low-temperature range

Answer 9: Thank you for pointing this out. I agree with this comment. The low-temperature range (-20℃--70℃) has been added in the revised manuscript (page 3, lines 99).

Comments 10: Line 103: MDCK cells, did not see the full name of MDCK in the manuscript, Madin-Darby canine kidney (MDCK) cells

Answer 10: Thank you for pointing this out. I agree with this comment. The full name of MDCK has been supplemented in the revised manuscript (page 3, lines 102).

Comments 11: Line 182: Unlike the diversity observed in the HA and NA genes, all 6 internal genes, of the isolated in this study belonged to the same lineage (Figures 2A-F).

Answer 11: Thank you for pointing this out. I agree with this comment. The sentence has been modified in the revised manuscript (page 4, line 181).

Comments 12: Line 225: HA comprises two subunits, HA1 and HA2, which are covalently linked by a disulfide bond[33]. A space needs to be created before these types of citations.

Answer 12: Thank you for pointing this out. I agree with this comment. Spaces have been added before all the citations that lacked spaces throughout the text.

Comments 13: Line 233: It is generally accepted that receptor-binding preference to human-type receptor receptors is the initial key step for a novel influenza-virus-causing-pandemic causing pandemic.

Answer 13: Thank you for pointing this out. I agree with this comment. The sentence has been modified in the revised manuscript (page 7, line 232-233).

Reviewer 2 Report

Comments and Suggestions for Authors

The swine influenza virus (SIV) is the etiology of swine influenza, causing severe acute respiratory disease and having significant impacts on animal husbandry and public health. In the manuscript entitled ‘Isolation and Characterization of H1 Subtype Swine Influenza Viruses Recently Circlating in China’, Yan et al. reported the isolation of 11 SIVs of H1 subtype from samples collected in China from 2021 to 2023. Their further investigation indicates that the recent H1 subtype SIVs circulating in China poses a potential threat to human health, and its is critical to continue monitoring their evolution and spread. These findings provide new insights into understanding the genetic and evolutionary charateristics of the recent circulating SIVs. Overall, the manuscript is well-organized. I have a few suggestions that may help improve the manuscript.

1.      Ln 13, this sentence is redundant because only one author has been marked with #.

2.      Ln 129, the statement ‘To assess the in vitro growth characteristics of the virus in different cells’ is inaccurate as only MDCK cells were used in this study.

3.      Ln 274, for ‘106 TCID50’, please use the upper and lower subscripts correctly.

Author Response

Comments and Suggestions for Authors

The swine influenza virus (SIV) is the etiology of swine influenza, causing severe acute respiratory disease and having significant impacts on animal husbandry and public health. In the manuscript entitled ‘Isolation and Characterization of H1 Subtype Swine Influenza Viruses Recently Circlating in China’, Yan et al. reported the isolation of 11 SIVs of H1 subtype from samples collected in China from 2021 to 2023. Their further investigation indicates that the recent H1 subtype SIVs circulating in China poses a potential threat to human health, and its is critical to continue monitoring their evolution and spread. These findings provide new insights into understanding the genetic and evolutionary charateristics of the recent circulating SIVs. Overall, the manuscript is well-organized. I have a few suggestions that may help improve the manuscript.

Thank you very much for taking the time to review this manuscript. Please find the detailed responses below and the corresponding revisions in the re-submitted files.

Comments 1: Ln 13, this sentence is redundant because only one author has been marked with #.

Answer 1: Thank you for pointing this out. I agree with this comment. The sentence has been removed in the revised manuscript.

Comments 2: Ln 129, the statement ‘To assess the in vitro growth characteristics of the virus in different cells’ is inaccurate as only MDCK cells were used in this study.

Answer 2: Thank you for pointing this out. I agree with this comment. In the revised manuscript, the sentence was modified to ‘To assess the in vitro growth characteristics of the virus’ (page 3, lines 128).

Comments 3: Ln 274, for ‘106 TCID50’, please use the upper and lower subscripts correctly.

Answer 3: Thank you for pointing this out. I agree with this comment. The upper and lower subscripts have been corrected in the revised manuscript (page 9, lines 273).